# Persistent Human Papillomavirus Infection

**DOI:** 10.3390/v13020321

**Published:** 2021-02-20

**Authors:** Ashley N. Della Fera, Alix Warburton, Tami L. Coursey, Simran Khurana, Alison A. McBride

**Affiliations:** Laboratory of Viral Diseases, Division of Intramural Research, National Institute of Allergy and Infectious Diseases, National Institutes of Health, Bethesda, MD 209892, USA; ashley.dellafera@nih.gov (A.N.D.F.); alix.warburton@nih.gov (A.W.); tami.coursey@nih.gov (T.L.C.); simran.khurana@nih.gov (S.K.)

**Keywords:** HPV, papillomavirus, persistence, extrachromosomal replication, tethering, cancer, epithelium, immune evasion, latency

## Abstract

**Simple Summary:**

The success of HPV as an infectious agent lies not within its ability to cause disease, but rather in the adeptness of the virus to establish long-term persistent infection. The ability of HPV to replicate and maintain its genome in a stratified epithelium is contingent on the manipulation of many host pathways. HPVs must abrogate host anti-viral defense programs, perturb the balance of cellular proliferation and differentiation, and hijack DNA damage signaling and repair pathways to replicate viral DNA in a stratified epithelium. Together, these characteristics contribute to the ability of HPV to achieve long-term and persistent infection and to its evolutionary success as an infectious agent.

**Abstract:**

Persistent infection with oncogenic human papillomavirus (HPV) types is responsible for ~5% of human cancers. The HPV infectious cycle can sustain long-term infection in stratified epithelia because viral DNA is maintained as low copy number extrachromosomal plasmids in the dividing basal cells of a lesion, while progeny viral genomes are amplified to large numbers in differentiated superficial cells. The viral E1 and E2 proteins initiate viral DNA replication and maintain and partition viral genomes, in concert with the cellular replication machinery. Additionally, the E5, E6, and E7 proteins are required to evade host immune responses and to produce a cellular environment that supports viral DNA replication. An unfortunate consequence of the manipulation of cellular proliferation and differentiation is that cells become at high risk for carcinogenesis.

## 1. Introduction

The *Papillomaviridae* family is comprised of a diverse group of ancient DNA viruses that are prevalent amongst a wide range of host species, including mammals, birds, reptiles, and fish. More than 650 distinct animal and human papillomavirus (HPV) types have been identified, and the 440 HPV types have co-evolved to exist and persist within the human population [http://pave.niaid.nih.gov/ (accessed on 16 February 2021)] [1]. The HPVs are classified phylogenetically according to DNA sequence homology in the L1 gene (which encodes the structural L1 capsid protein) into five genera: *Alpha, Beta, Gamma, Mu*, and *Nu* [2]. The viruses that comprise each genus have evolved to adapt to distinct ecological niches within their host. Specifically, viruses within the *Beta, Gamma, Mu*, and *Nu* genera infect the cutaneous epithelium, whereas viruses within the *Alpha* genus infect both cutaneous and mucosal epithelia. In addition to this distinct tissue tropism, the viruses within these genera differ in their associations with clinical disease [3]. HPVs from the *Beta* and *Gamma* genera cause primarily asymptomatic infections [3]. The mucosal *Alpha* HPV types are further classified as low-risk HPVs (LR-HPVs) or high-risk HPVs (HR-HPVs) based on their ability to cause cancer [3]. HR-HPV infection is the causative agent of almost all cases of cervical cancer in women and is also highly associated with cancers of the lower genital tract, anus, and oropharynx in both men and women [4]. Anogenital infection with the oncogenic HR-HPVs is very common and most infections are managed immunologically and cleared in a period of one to two years [5]. The development of HPV-mediated cancer is associated with long-term persistent infection; continual expression of the viral oncogenic E6 and E7 proteins abrogates cell cycle checkpoints and inhibits immune detection. Consequently, the infected cells over proliferate and cellular mutations accumulate, leading to the formation of HPV-associated cancers [4].

In this review, we will describe the factors that promote persistent infection by the oncogenic HR-HPVs.

## 2. Natural History of HPV Infection

Infection with the various HPV types gives rise to a spectrum of subclinical and clinical manifestations, ranging anywhere from asymptomatic infection to benign warts or papillomas on the skin and genitalia. While many HPVs can be considered commensal and part of the microbiota of healthy tissue, long-term persistent infection with HR-HPVs increases the risk for oncogenic progression and can lead to invasive cancer. As such, HR-HPV infection is the causative agent of almost all cases of cervical cancer in women and is also highly associated with cancers of the lower genital tract, anus, and oropharynx in both men and women [4]. Despite the introduction of the HPV vaccine in 2006, low vaccine uptake and vaccine hesitancy [6] will result in the continued occurrence of HPV-associated cancers in both men and women [7]. Therefore, HPV infection will remain a significant health burden in the upcoming years.

Individual HPV types infect and replicate in keratinocytes located within the cutaneous and mucosal epithelia. While the squamous epithelia of these tissue surfaces are vulnerable to infection, the outcome of infection is likely contingent on the inherent properties of the originally infected cell. This is particularly true for HPV-associated cancers of the cervix, oropharynx, and anus. Specifically, cells with increased susceptibility to infection and oncogenesis reside within cellular transition zones of these defined anatomical regions. These locations are more infection prone as the accessibility of the proliferative basal cells is increased where the junctions of two epithelial cell types meet [8,9]. Recent studies also support a potential role for stem cells in viral persistence and oncogenesis [10]. Infection of a stem cell is much more likely to give rise to long-term infection than that of a transit amplifying cell [11].

Natural HPV infection rarely persists longer than two years and over 90% of detectable infections are resolved and not detected within five to seven years [4]. While innate immune responses are typically able to clear incident infections early on, the propensity of an established HPV lesion to regress depends on a robust cell-mediated response [4]. As such, persistent HPV infection occurs in those individuals who are unable to mount the appropriate innate and adaptive immune responses.

The usual course of cervical HPV infection is initial acquisition, persistence of infection, and less frequently, neoplastic progression (Figure 1). At early stages, infections are cleared by the immune system and neoplasias can regress. However, it is still not clear whether all infected individuals seroconvert and whether they are then resistant to infection by the same HPV type [12]. The incidence of HPV infection peaks in women in their early 20s and then becomes undetectable [4]. However, a small number of women have newly detectable infection in middle age and it is not clear whether this is due to a new infection, or reactivation of a latent infection [12]. Latency has been observed experimentally in rabbit models of papillomavirus infection [13,14], and focal regions of silent infection have been detected in the human cervix [15]. Therefore, HPV latency could increase the likelihood of persistent infection in the human population.

## 3. HPV Genome

All HR-HPVs have small circular double stranded DNA genomes that are 7–8 kb in size and encode two sets of conserved core proteins; those required for viral DNA replication, E1 and E2, and the structural proteins essential for virion assembly, L1 and L2. Additionally, the HR-HPV genomes encode accessory genes, E4, E5, E6, and E7, that modify the host epithelium to create an environment suitable for viral replication and to facilitate evasion of innate immune responses [16]. Together, these genes function to establish persistent HPV infection and to support the productive HPV life cycle. A generic oncogenic human *Alphapapillomavirus* genome is shown in Figure 2.

The HPV genome is organized into three regions, two coding regions that encode the viral proteins and a non-coding region that regulates viral transcription and replication. The coding regions of the viral genome contain between seven and nine open reading frames that are organized into early and late regions; the early region encodes the E1, E2, E1^E4, E8^E2, E5, E6, and E7 proteins and the late region encodes the L1 and L2 capsid proteins. The non-coding region, otherwise known as the upstream regulatory region (URR), is located upstream of the early region and contains multiple cis regulatory elements required for transcription, as well as the origin of replication [17]. Transcription from the viral genome occurs in three phases (early, intermediate, and late) and is intricately linked to the host epithelial differentiation program. Early viral gene transcription is initiated from the P_E_ early promoter in undifferentiated basal keratinocytes and is terminated at the pA_E_ early polyadenylation site. In suprabasal cells, intermediate transcription is initiated from the P_L_ late promoter and is terminated at the early polyadenylation site. This results in increased levels of the E1 and E2 replication proteins necessary for DNA amplification. Finally, late viral transcription is initiated from the P_L_ late promoter, and is terminated at the pA_L_ late polyadenylation site, resulting in expression of the structural L1 and L2 capsid proteins [18].

## 4. Overview of the HPV Life Cycle

The HPV life cycle exploits the host differentiation program of the stratified cutaneous and mucosal epithelia for productive infection. A schematic of the HPV infectious cycle and pattern of viral gene expression is shown in Figure 3. To initiate infection, the HPV viral particle must first access the dividing basal cells of the lower epithelium through a micro-abrasion or wound in the stratified epithelium. Attachment of the viral particle to heparan sulfate proteoglycans on the basement membrane, followed by transfer to an uncharacterized secondary receptor on keratinocytes, induces a series of conformational changes that promote viral entry [19].

The HPV particle enters basal keratinocyte cells by endocytosis, during which the L2 protein inserts into the membrane and cloaks the virus in a membrane vesicle [20]. L2 subsequently associates with cytoplasmic trafficking factors to facilitate transport of HPV to the trans golgi network, where the HPV-containing vesicle resides until gaining access to the host nucleus [21]. Following breakdown of the nuclear envelope during mitosis [22], the vesicle enters the nucleus and, through L2, associates with condensed mitotic chromosomes [21,23]. A central region in the L2 minor capsid protein facilitates viral genome tethering [23]. The HPV-harboring vesicle remains associated with the host chromosomes until mitosis is finished and the nuclear envelope is restored.

After nuclear delivery, the viral DNA localizes to promyelocytic leukemia nuclear bodies (PML-NBs) [24], and it is likely that viral transcription and DNA replication initiate there. Despite the role of host PML-NBs in limiting gene expression of most DNA viruses, efficient transcription of the HPV genome is reliant on PML-NBs and displacement of the PML-NB resident protein Sp100 [24]. As such, HPV utilizes some components of PML-NBs, while evading others to ensure successful infection.

The HPV genome utilizes three phases of viral DNA replication: initial amplification, maintenance, and vegetative amplification. During initial amplification, the viral genome goes through a few rounds of DNA replication to establish a small number of extrachromosomal genomes that will persist in the self-renewing basal cells of the lower epithelium. This pool of infected basal cells is the foundation of the infected lesion and serves as a reservoir for persistent HPV infection. Maintenance replication occurs within the proliferating cells, and there, the extrachromosomal viral genomes replicate along with the host cellular DNA and are tethered to host chromatin to ensure partitioning into daughter cells. Viral copy number is maintained at a constant number until the cells begin to differentiate and move towards the surface of the epithelium, at which point late viral gene expression and high levels of DNA amplification occur (Figure 4).

During differentiation of keratinocytes, the late promoter located in the E7 gene is activated, and high levels of the E1 and E2 proteins and late gene products, including E1^E4, L1, and L2, are expressed [18]. Expression of late transcripts that include L1 and L2 depend on alternate polyadenylation site recognition and alternative splicing mechanisms [18]. Finally, the viral genome is packaged and viral progeny are shed in squames from the surface of the epithelium [3].

## 5. Creating an Environment Conducive to Persistent Viral Replication

All papillomaviruses have developed strategies to facilitate viral genome replication and maintenance within the stratified epithelia. The HR-HPVs use the viral accessory proteins E5, E6, and E7 to adapt to specific ecological niches, to establish a cellular environment favorable for viral replication and persistence, and to evade host immunosurveillance programs [25], as illustrated in Figure 5. Specifically, these proteins modify the cellular environment to promote persistence within the proliferative basal keratinocytes and to support vegetative replication within the terminally differentiated keratinocytes. The E5, E6, and E7 accessory proteins also safeguard the viral genome from host innate and adaptive immune responses throughout the infectious cycle. In fact, it has been proposed that the E6 and E7 activities of the HR-HPVs that promote oncogenesis originally evolved to inactivate cellular pathways required for immune evasion [26].

### 5.1. E5, E6, and E7 Proteins Regulate Cellular Proliferation and Differentiation

The capacity of HR-HPVs to persist within the self-renewing cells of the host epithelium relies on a vast array of interactions between the viral E5, E6, and E7 proteins and a multitude of cellular proteins (Figure 5). These interactions promote proliferation and cell cycle progression in cells that normally would have exited the cell cycle. Manipulation of the host cellular environment in this way ensures that the virus has the necessary factors required for DNA replication.

The E5, E6, and E7 proteins alter the basal and parabasal epithelial layers of the squamous epithelium by inducing cell proliferation and delaying differentiation. Moreover, E6 and E7 also drive cell cycle progression and viral genome amplification in the postmitotic, differentiated cells in the upper epithelial layers. The E5 protein increases cell proliferation and inhibits keratinocyte differentiation by augmenting epidermal growth factor receptor (EGFR) signaling [27]. The E6 and E7 proteins regulate cell-cycle entry, DNA synthesis, long term cell division, keratinocyte differentiation and apoptosis by inactivating the host p53 and retinoblastoma protein (pRb) tumor suppressor pathways [4]. The Rb protein family regulates the G1–S phase transition of the cell cycle by modulating the activity of E2F transcription factors. During HR-HPV infection, the E7 protein binds and degrades pRb [28,29], leading to S-phase entry and host DNA synthesis. E7 expression results in transcriptional activation of the KDM6A and KDM6B histone demethylases, which induce global demethylation of histone H3K27 and epigenetic reprogramming of HPV-positive cells [30]. Global reduction of H3K27 methylation results in increased expression of the tumor suppressor p16^INK4A^, which is a mediator of oncogene induced senescence and a biomarker for HR-HPV infection [31]. However, HR-HPV E7 abrogates oncogene induced senescence by inactivation of the pRb pathway.

Abrogation of the pRb-E2F pathway by E7 results in increased expression of p53; this is counteracted by the E6 protein, which instigates the ubiquitin-mediated degradation of p53 through its interaction with the E3 ubiquitin ligase E6-associated protein (E6AP) [32]. As continual cell division erodes telomeres and results in cellular senescence, the E6 protein also transcriptionally activates human telomerase reverse transcriptase (hTERT) [33], which stabilizes the telomere ends and prevents replicative senescence.

In addition, the HR-HPV E6 proteins bind and degrade members of the postsynaptic density protein, disc large tumor suppressor, zonula occludens-1 domain-containing proteins (PDZ) protein family through interaction with a PDZ binding motif in the C-terminus of E6 [34]. PDZ domain-containing proteins regulate epithelial cell polarity and asymmetric cell division and disruption of these processes promotes the infectious life cycle. Of note, E6 and E7 expression is required for long-term maintenance of viral genomes in keratinocytes [35,36,37]. While the E6–PDZ interaction is important for ensuring persistence of HPV genomes [38], the E6-mediated inactivation of p53 is essential [35,39].

### 5.2. E5, E6, and E7 Proteins Impede Various Steps of the Innate and Adaptive Immune Responses

The success of HPV as an infectious agent lies within the ability of the virus to evade host immune responses and establish long-term persistent infection. The infectious cycle of HPV itself provides barriers to host immune recognition. High levels of viral gene expression and vegetative amplification occur in differentiated keratinocytes that are eventually sloughed from the surface of the epithelium by the process of terminal differentiation. As such, these viral processes are hidden from host immunosurveillance. Additionally, the virus does not induce cytolytic death and does not stimulate inflammation and expression of proinflammatory cytokines. These cytokines would normally recruit immune effector cells that are required for antigen presentation and clearance of virally infected cells. Collectively, these strategies of viral infection shield the virus from host innate and adaptive defenses for long and variable periods of time [40].

Host innate immune defenses against viral infection consist of three fundamental activities: detection of foreign viral nucleic acids, activation of various signal transduction pathways, and production of proinflammatory and anti-viral cytokines. HR-HPVs have developed several strategies to evade host immunosurveillance programs and to perturb innate immune signaling pathways. Specifically, E5, E6, and E7 inhibit host DNA sensors, interfere with cytokine production, obstruct the activation of signaling pathways, and regulate proinflammatory responses [41,42]. E7 inhibits the first point of the cellular signaling cascade by binding to the cytosolic DNA sensor stimulator of interferon genes (STING) and inhibiting subsequent downstream production of type I interferons (IFNs). E5, E6, and E7 proteins hinder type I IFN and proinflammatory signal transduction pathways to impede Janus kinase/signal transducer and activator of transcription (JAK/STAT) and nuclear factor κB (NF-κB) signaling [41,43]. Moreover, the E5 protein downregulates the keratinocyte specific IFN, IFN-K [41]. Finally, E6 and E7 impair the formation of host inflammasome complexes by reducing recruitment of members of this complex to sites of viral infection and lowering expression of the secreted form of proinflammatory IL-1, IL-1B. These mechanisms, along with low expression of innate immune response factors in the tissues where HPV activity is the highest, contribute to HPV’s success as an infectious agent [44].

Many viruses have evolved mechanisms to avoid restriction of viral replication caused by apolipoprotein B mRNA editing enzyme, catalytic polypeptide-like 3 (APOBEC3) mutagenesis [45]. Paradoxically, the HR-HPVs induce APOBEC3 expression and it has been proposed that inactivation of the pRB pathway allows expression of endogenous retrotransposons, which in turn induce APOBEC3 expression [45]. HPV genomes are somewhat depleted of APOBEC3 target sequences indicating that APOBEC3 has contributed to the evolution of HPV genomes [46,47]. Furthermore, persistent cervical infections containing HPV genomes with APOBEC3 mutational signatures are more likely to be cleared [47]. Thus, it is likely that APOBEC3 enzymes engage with the HPV genome during infection but it is not clear whether HPVs counteract this response or use it to their advantage [45].

While the innate immune response to viral infection lacks explicit memory, this first line of defense is crucial to activate an adequate adaptive immune response. The role of the cell-mediated adaptive immune response is to recognize foreign antigens presented by antigen presenting cells (APCs) and to destroy and clear virally infected cells. The E5, E6, and E7 proteins perturb this in three ways; HPV impedes the recruitment of epidermal APCs, decreases viral antigen uptake in APCs, and downregulates the expression of the antigen presenting major histocompatibility complex I (MHC I) molecules on the surface of HPV infected keratinocytes [48,49]. This reduction in surface MHC I molecules helps avoid recognition of infected cells by cytotoxic T lymphocytes and thus facilitates persistent infection [49]. The numerous strategies employed by the HR-HPVs to avoid these innate, and adaptive responses allows the virus to persist within the host keratinocytes and remain undetected for long periods of time.

## 6. Papillomavirus Genome Replication

### 6.1. Initial Stage of Replication

Central to the success of persistent infection is the ability of HPV to replicate and maintain its genome by hijacking various cellular processes at different stages of the infectious cycle. Papillomaviruses access the basal layer of the stratified epithelium through microabrasions and upon entry into the nucleus the E1 and E2 proteins are expressed and direct low-level amplification of the viral genome. E1 is an ATP-dependent helicase that binds to and unwinds the viral replication origin. E2 is the primary transcriptional regulator of the virus, but also functions during replication as the helicase loader. Together, E1 and E2 cooperatively bind to the replication origin to initiate viral DNA synthesis [50]. The minimal replication origin consists of an E1 binding site (a cluster of overlapping palindromic recognition sequences), flanked by E2 binding sites (E2BS) (see Figure 6).

The E1 protein has four functional domains, an N-terminal domain that regulates nucleocytoplasmic transport, a sequence specific DNA binding domain, an oligomerization domain that induces hexamer formation, and the C-terminal helicase [51]. E2 is composed of two conserved domains connected by a flexible hinge: an N-terminal transcriptional activation domain and a sequence specific C-terminal DNA binding domain (Figure 6A). The E2 protein exists as a dimer and dimerization is mediated through the DNA binding domain. The unstructured hinge is divergent among papillomavirus species and unnecessary for E2 transcription or replication activities, but post-translational modifications of this region can regulate E2 nuclear localization, half-life, protein–protein interactions, and host chromatin tethering [52].

To initiate DNA synthesis, E1 and E2 co-operatively bind to the replication origin; the DNA binding domain of each protein associates with its respective binding site and the N-terminal domain of E2 interacts with the E1 helicase domain to form a high affinity origin recognition complex [53]. Next, a conformational change displaces E2 from the complex and additional E1 proteins are recruited to form two intermediate E1 trimers at the origin [54]. Finally, the DNA at the origin is unwound bidirectionally by double-hexameric E1 helicases [55] and cellular proteins are recruited to synthesize DNA in a theta-mode [51] similar to other small circular DNA viruses.

### 6.2. Establishment

After initial HPV DNA amplification, the viral genomes must be “established” in the host cell nucleus as stable, low copy number extrachromosomal elements. These genomes must evade detection by innate immune defenses and associate with transcriptionally active regions of host chromatin, to avoid being silenced [56]. Successful establishment is most likely a rare event, but it culminates with viral genomes being associated with favorable regions of host chromatin.

### 6.3. Maintenance Replication

One hallmark of papillomavirus replication is that the genomes are replicated and maintained as extrachromosomal plasmids in cell lines derived from cervical lesions, or in keratinocytes immortalized with HR-HPV genomes. The viral genome is replicated in S-phase concomitant with cellular DNA replication, and the copy number remains stable over many cell divisions. This replication mode is thought to represent the maintenance replication that occurs in the lower, dividing cells of an HPV infected lesion.

The requirements for maintenance replication were first elucidated for bovine papillomavirus type 1 (BPV1) [57]. This showed that the E1 and E2 proteins supported only transient replication of plasmids containing the minimal replication origin, while long term maintenance replication required, in addition, regions of DNA from the URR that contained multiple E2 binding sites. This was the first indication that the E2 protein had an additional role in maintenance replication, and the subsequent discovery that both the E2 protein and viral genomes were associated with mitotic chromosomes resulted in the hitchhiking model [58]. In this model, the E2 protein binds to binding sites in the viral genome and tethers the genomes to host chromosomes by protein–protein interactions with the transactivation domain [59]. Consequently, genomes are partitioned in approximately equal numbers to daughter cells. This is an appealing strategy, and similar to that used by the *Gammaherpesviruses* Epstein Barr virus and Kaposi’s sarcoma herpesvirus, but the exact mechanism is complex, and many details still need to be determined [60].

In BPV1, six E2 binding sites contained within the enhancer region of the URR form a minichromosome maintenance element (MME). This MME is required in addition to the minimal replication origin for maintenance replication [57], and can be substituted by six oligomerized E2BSs. In contrast, the HR-HPVs contain only four E2BS, three of which are in the replication origin (Figure 6B). In a plasmid retention assay, two HPV18 E2BS from the origin were sufficient to maintain non-replicating plasmids in dividing cells [61], further supporting the role of the E2 protein in this process. In keratinocytes, HPV18-derived replicons complemented in trans by the entire HPV18 genome required only the minimal replication origin (with three E2BS) and sequences from the transcriptional enhancer (designated the minichromosome maintenance enhancer element (MMEE) [62]. Therefore, while the E2 tethering model still holds, it is likely mediated by complexes of the viral E2 protein and cellular factors associated with the viral genome. Recent studies have indicated that cis elements outside of the papillomavirus URR might also be important for maintenance replication. For example, CCCTC-binding factor (CTCF) binding sites located in the late region of HPV31 promote maintenance replication in dividing cells [63].

Much work has sought to find the target region or target protein of the tethering complex on host chromosomes. The E2 proteins from many papillomaviruses are observed in punctate speckles on host chromosomes (e.g., BPV1, HPV E2 proteins from *Betapapillomaviruses* and *Mupapillomaviruses*) while E2 proteins from the mucosal *Alphapapillomaviruses* (LR and HR-HPVs) are difficult to detect on chromosomes by immunofluorescence. Multiple cellular factors may contribute to the tethering complex, but the double bromodomain factor, bromodomain-containing protein 4 (Brd4), is the most thoroughly investigated candidate. Brd4 interacts with all E2 proteins to regulate viral transcription [64,65]. In complex with E2, Brd4 represses papillomavirus transcription from the early promoter [66,67], but promotes early transcription in the absence of E2 [68]. Brd4 also colocalizes with those E2 proteins that can be readily detected on mitotic chromosomes [64,65,69]. However, Brd4 interactions with *Alphapapillomaviruses* are weak and difficult to observe on mitotic chromosomes except in late telophase or under specific fixation conditions [65,70]. Furthermore, HPV31 genomes with a mutation in E2 that disrupts the interaction of the N-terminal domain of E2 with the C-terminus of Brd4 can replicate persistently [71]. However, the determinants of the interactions of HPV E2s and Brd4 are complex [72], and the involvement of Brd4 in many viral processes makes it difficult to determine the precise role of Brd4 in HPV tethering. Additional candidates proposed to influence genome maintenance include structural maintenance of chromosome (SMC) architectural proteins, TopBP1, CTCF, and the DNA helicase ChlR1 [60].

E2 proteins with a high affinity for Brd4 (e.g., HPV1 E2), colocalize with Brd4 as punctate speckles on mitotic chromosomes [69] and these sites were mapped by Brd4–E2 ChIP-chip (chromatin immunoprecipitation-chip microarray) [73]. These sites were shown to be enriched in chromatin acetylation and methylation associated with transcriptionally active chromatin, but also to have many characteristics of common fragile sites. This led to the hypothesis that E2–Brd4 complexes associated the HPV genomes with regions of host chromatin susceptible to replication stress and that this might be advantageous for productive HPV infection in differentiating cells when the viral genome switches to a recombination-dependent replication mode that relies of the DNA damage response [74,75]. In situ Hi-C analysis of viral and host DNA interactions in cell lines with extrachromosomal HPV16 and HPV31 further demonstrated that HPV genomes were tethered to euchromatic, gene-rich regions of chromatin [76].

There are several ways in which extrachromosomal viral genomes can be partitioned to daughter cells and this has been best studied for the *Gammaherpesviruses* [77]. Viral genomes can associate with mitotic chromosomes before S-phase with each daughter molecule being faithfully segregated to a daughter cell in association with the newly replicated chromosome. Alternatively, viral genomes can randomly associate with host chromosomes after DNA replication of both the virus and host, with approximately equal numbers of genomes being partitioned to daughter cells. The association of HPV DNA with host chromosomes at all stages of the cell cycle [76] suggests that HPV is faithfully partitioned, but this must still be formally proven.

In general, it is thought that the E1 and E2 proteins initiate viral DNA synthesis at the replication origin in all modes of replication and additionally that E2 facilitates genome partitioning in the maintenance phase. However, there is evidence that E1 might not always be required for extrachromosomal replication during the maintenance phase [78,79]. The E1 protein can induce DNA damage and, concomitantly, the DNA damage response (DDR) and thus is detrimental to cellular proliferation [80,81,82]. Nucleocytoplasmic transport of E1 is regulated by phosphorylation of the N-terminal domain, to ensure that E1 is retained in the cytoplasm except during S-phase replication [81]. Therefore, it would be advantageous in some circumstances for the cellular replication machinery to initiate DNA synthesis of the viral genome in the absence of E1.

Cellular DNA is licensed so that it only undergoes one round of replication per cell cycle. It has long been debated whether papillomavirus genomes are similarly restricted in the maintenance phase or whether they replicate by a random choice mechanism (some genomes remaining unreplicated while others undergo multiple rounds of replication). Taking the evidence together, it seems most plausible that E1-initiated replication undergoes unlicensed replication and that E1-independent replication is licensed [83]. In this scenario, DNA amplification at the early and late stages of infection would undergo unlicensed replication and replication could be either licensed or unlicensed in the maintenance phase. In cell culture, viral genomes can amplify when cells reach confluence, complicating analyses of genome licensing [84,85].

All papillomaviruses encode a truncated E2 protein, E8^E2, that modulates viral transcription and replication to maintain the persistent maintenance phase of the life cycle [86]. The E8^E2 repressor protein consists of a short E8 peptide fused to the hinge and DNA binding domain of the E2 proteins, as shown in Figure 6A [86]. The E8^E2 protein can compete with full-length E2 for binding to the E2BSs to suppress E2 dependent replication and transcriptional regulation, but E8^E2 can also repress by forming heterodimers with the full-length E2 protein [86]. Additionally, the E8 moiety of the repressor protein recruits the host nuclear corepressor complex (NCOR/SMRT) to the HPV genomes [87]. Without E8^E2, HPV genomes spontaneously enter the productive phase of the life cycle, inducing the cellular DDR and inhibiting cell growth [88]. Thus, the E8^E2 protein is vital for persistent HPV infection by repressing the productive phase, thereby maintaining the reservoir of replicating viral genomes in the basal cells of a lesion.

### 6.4. Differentiation-Dependent Viral DNA Amplification

Vegetative DNA amplification of the HPV genome occurs in differentiated suprabasal cells in the upper layers of the stratified epithelium. These terminally differentiated cells have exited S-phase and no longer contain the replicative factors required to synthesize DNA. As such, HPV induces host DNA damage signaling and hijacks the cellular DDR repair machinery as a means to acquire enzymes essential for viral DNA replication in the G2 arrested cell [89]. Utilization of the host DDR pathway is beneficial to HPV in two ways: HPV can replicate in terminally differentiated cells, and there is no competition with the host cell for factors required for DNA synthesis. This replication strategy promotes persistent low-level DNA replication in the basal cells while simultaneously generating progeny virions in the terminally differentiated layers of a lesion.

Amplification of the HPV genome takes place in defined nuclear compartments known as replication factories. These factories mimic endogenous DNA damage foci or collapsed replication forks and play an important role in activating signaling pathways that induce and recruit an influx of repair proteins to sites of viral replication [90]. Various components of the host DDR signaling pathway including pATM, pATR, γH2AX, pChk2, pChk1, BRCA1, RAD51, TopBP1, and pNBS1 are recruited to and concentrated in these factories [74,80,91,92,93,94,95]. Moreover, numerous DDR associated histone acetyltransferases and deacetylases promote this late stage of replication; both acetyltransferase TIP60 [42] and the deacetylase, SIRT1, promote productive replication and late gene expression [96,97]. The HR-HPV E7 oncoprotein activates the host ataxia teleangiectasia mutated (ATM) kinase signaling pathway, which is essential for viral DNA amplification [75]. E7 also hijacks the E3 ubiquitin–protein ligase RNF168 by blocking its function at cellular double strand breaks to promote homologous replication at viral replication centers [98]. Thus, E7 promotes the accumulation of DNA repair factors at the replication foci to facilitate differentiation dependent genome amplification and processing.

The exact mechanism of vegetative DNA replication is not well elucidated, but the recruitment of homologous recombination factors to the viral replication foci suggests that replication switches to a recombination dependent replication mode during vegetative amplification phase to produce high copy numbers of the viral genome and to facilitate high fidelity viral DNA synthesis [74,99].

## 7. Consequences of Persistent Infection with HR-HPVs

One consequence of persistent HR-HPV infection is the accidental integration of the viral genome into host chromatin [16]. Integration is common in HPV-associated cancers, but is a dead-end for the virus as they can no longer complete the lifecycle and synthesize infectious virions. There are several mechanisms by which integration can promote oncogenesis, and these are in part dependent on where the integration breakpoint occurs in the viral genome. For example, integration in the E2 gene disrupts E2-mediated repression of the early promoter and subsequent dysregulation of E6 and E7 gene expression. This results in increased cellular proliferation, inhibition of cell-cycle checkpoints, and progressive genetic instability [100]. Integration can also be associated with host genome amplifications and/or rearrangements that promote oncogenesis. Integrated HPV genomes are detected in >80% of HPV-positive cervical tumors [101]; HPV18 is integrated in 100% of related cancers while HPV16 is only integrated in ~74% of cases [101,102]. In HPV-positive oropharyngeal carcinomas, the frequency of integration is lower than in cervical cancer, and the HPV genome is either extrachromosomal, integrated or a combination of both. It has been proposed that in some cases the extrachromosomal DNA is composed of viral-host hybrid DNA molecules [103].

Most HPV-associated cancers result from the undesirable effects of the viral oncoproteins in rewiring host cells to support persistent infections. Throughout the viral lifecycle, papillomavirus genomes associate with regions of transcriptionally active host chromatin to facilitate viral transcription, replication, persistence, and DNA amplification; notably, integration often occurs in these regions [104,105]. HPV integration also frequently occurs in common fragile sites [106,107], which are regions of the genome susceptible to replication stress. Viral replication occurs proximal to fragile sites and could greatly increase the chances of integration at these genetically unstable regions [73].

## Figures and Tables

**Figure 1 viruses-13-00321-f001:**
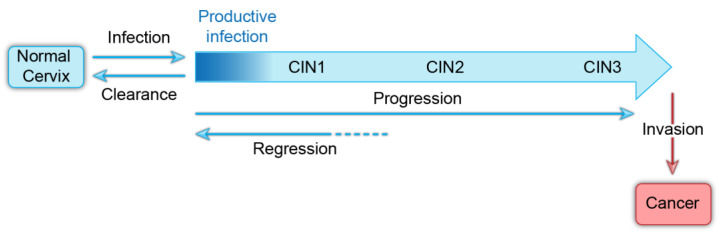
Natural history of oncogenic human papillomavirus infection. A model showing the progression of HPV infection to invasive cancer. Infection with HPVs is usually cleared by the immune system within a couple of years. Persistently infected cells can regress, but over time can progress to invasive cancer.

**Figure 2 viruses-13-00321-f002:**
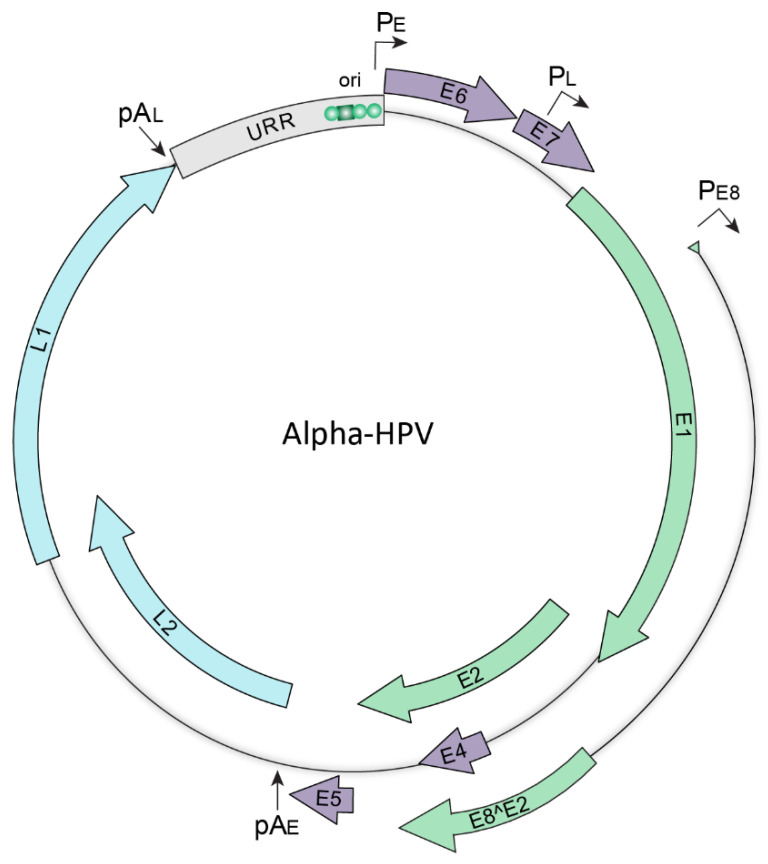
Map of the HPV genome. Schematic representation of an *Alphapapillomavirus* genome. The green, purple, and blue arrows represent the early, accessory, and late viral open reading frames, respectively. The upstream regulatory region (URR) shown in grey contains regulatory elements including the origin of replication (ori) that contains binding sites for the E1 and E2 replication proteins (denoted by a green square and circles, respectively). The early (P_E_), late (P_L_) and E8 (P_E8_) promoters, and the early (pA_E_) and late (pA_L_) polyadenylation sites are indicated.

**Figure 3 viruses-13-00321-f003:**
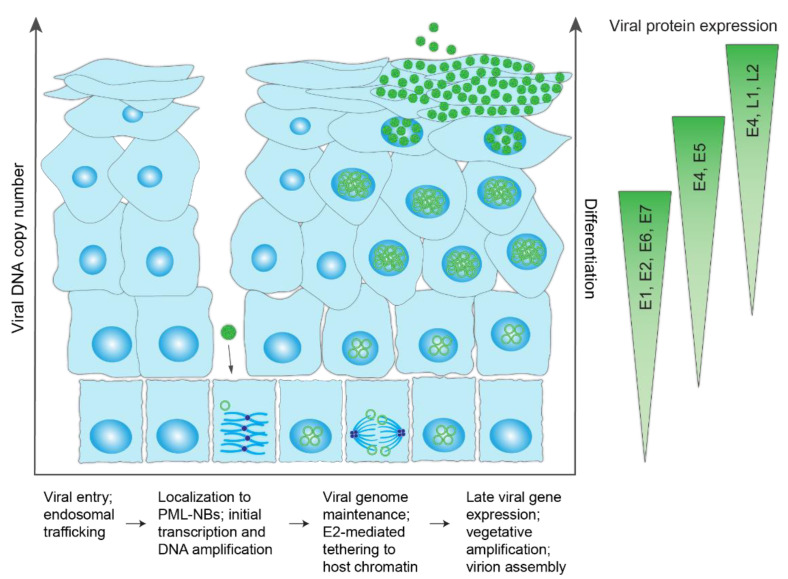
HPV infectious life cycle. Schematic representation of the differentiated layers of a stratified epithelium infected with HPV. The virus accesses the basal keratinocytes through a microabrasion. Upon cellular entry, the virus is trafficked through the endosome and enters the nucleus (encased in a membrane vesicle) following breakdown of the nuclear membrane during mitosis. Within the nucleus, HPV genomes localize to promyelocytic leukemia nuclear bodies (PML-NBs), undergo a limited round of DNA synthesis and become established by tethering to host chromatin to maintain the viral genome at a constant copy number in dividing cells. Upon epithelial differentiation, infected cells amplify the viral DNA to high copy numbers, and late viral genes are expressed for virion assembly/packaging. Virions are sloughed from the epithelial surface in viral-laden squames. The different steps in the viral lifecycle are summarized below the schematic and the corresponding viral protein expression levels indicated on the right.

**Figure 4 viruses-13-00321-f004:**
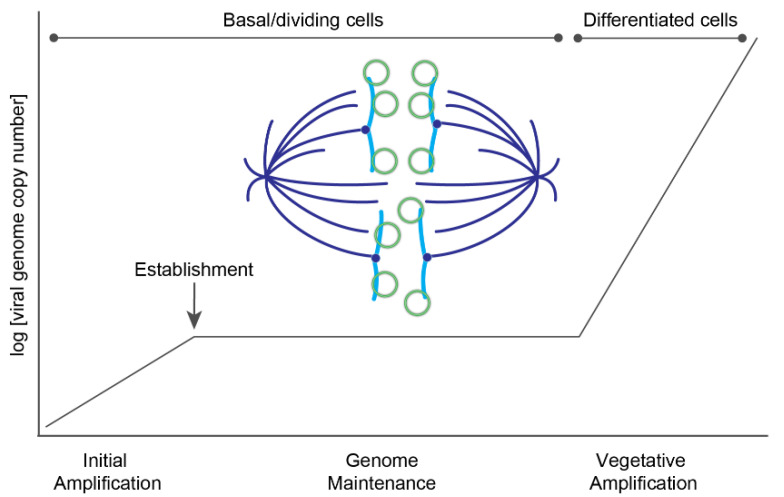
Phases of HPV replication. A plot showing the different phases of HPV replication in the host epithelium. Upon entry, HPV genomes undergo a limited burst of DNA amplification. The viral genome becomes established in the nucleus, and in the maintenance phase is replicated at a low copy number and partitioned to daughter cells. In differentiated cells, the viral DNA undergoes a second burst of amplification to a very high copy number to generate genomes for progeny virions. A model of viral partitioning during the maintenance phase is illustrated; HPV genomes (green circles) attach to host chromosomes (blue) and are partitioned to daughter cells during mitosis. The mitotic spindle is shown in dark blue.

**Figure 5 viruses-13-00321-f005:**
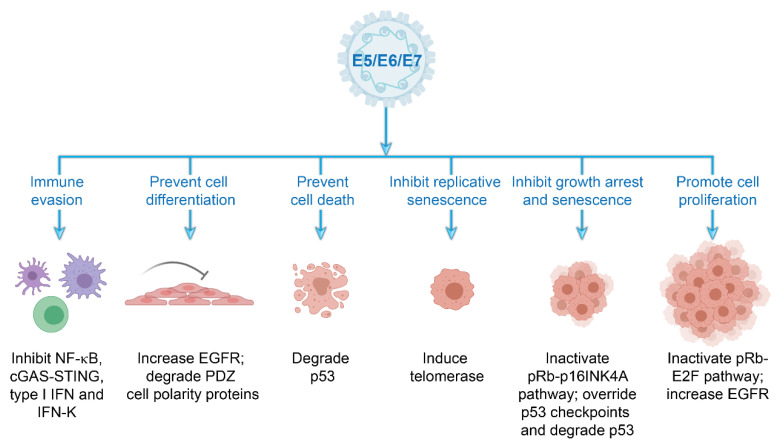
Cellular processes modulated by the high-risk HPV accessory proteins. Schematic representation of the multiple strategies employed by the E5, E6, and E7 accessory proteins to establish a cellular environment in tissue-specific niches that supports viral replication and persistence and evade immune surveillance. Abbreviations: cGAS–STING, cyclic GMP–AMP synthase–stimulator of interferon genes; EGFR, epidermal growth factor receptor; IFN, interferon; NF-κB, nuclear factor κB, PDZ, postsynaptic density protein, disc large tumor suppressor, zonula occludens-1 domain-containing proteins; pRb, retinoblastoma protein. Images created with BioRender.com.

**Figure 6 viruses-13-00321-f006:**
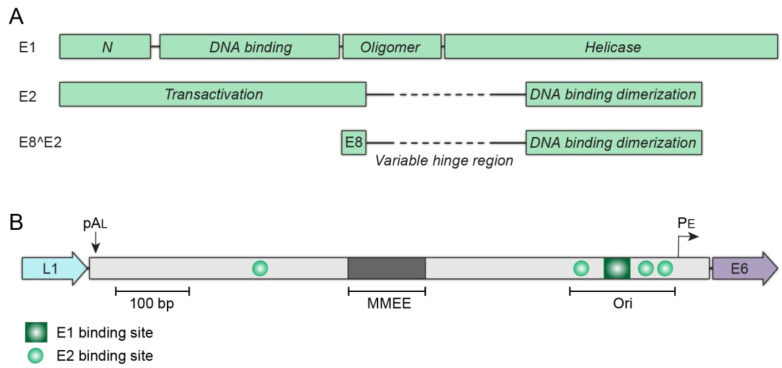
Replication proteins and elements. (**A**) The structural and functional domains of the E1, E2, and E8^E2 proteins. (**B**) The upstream regulatory region (URR) of the *Alphapapillomavirus* HPV18, illustrating the cis-elements required for stable viral genome maintenance. The minichromosome maintenance enhancer element (MMEE) and origin of replication (ori) are indicated. The E2 binding sites are represented by green circles and the E1 binding site by a green square. The late polyadenylation site pAL and early viral promoter P_E_ are also shown.

## Data Availability

Not applicable.

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
