# Peer review of "Persistent Human Papillomavirus Infection"

_viruses, 2021, doi:10.3390/v13020321_

Round 1

Reviewer 1 Report

A brilliant and very educational review article that I enjoyed reading. I believe the scientific community will find it useful. My only minor comment, since I have been working in this field myself, is the lack of mention of the mutational imprint on replicated HPV genomes left by the anti viral cytidine deaminase/APOBEC-pathways. A very brief mention of the differences in oncogenic transformation observed between HPV16 and HPV18 in regard to integration may also be considered.

Author Response

A brilliant and very educational review article that I enjoyed reading. I believe the scientific community will find it useful. My only minor comment, since I have been working in this field myself, is the lack of mention of the mutational imprint on replicated HPV genomes left by the anti viral cytidine deaminase/APOBEC-pathways. A very brief mention of the differences in oncogenic transformation observed between HPV16 and HPV18 in regard to integration may also be considered. 

  • We have added information about the role of APOBEC on HPV infection and the resulting mutational imprint (lines 342-352)
  • We have added some more information about the differences observed in the frequencies of HPV16 and HPV18 integration (lines 584-585)

Reviewer 2 Report

The authors reviewed and summarized the current literature of the oncogenic human papillomavirus (HPV) types, their related genome structure and gene expression regulation. The authors also depicted the ability of HPV to hide from the immune system and therefore stay persistent for years, which then can lead to cancer development.

The review is presented in an easy-to-read and comprehensible manner. This topic is of (clinical) relevance – not only for virologists - and is in the scope of the journal. It can support the interested reader to understand the HPV pathogenesis in more detail.

However, I would propose extending or editing some paragraphs:

  1. Page 6, Line 15: As E6 and E7 are the key players in cancer development through p53 degradation and cell cycle activation via Rb I would suggest extending this section and consider including a schematic figure explaining those pathways. An illustration could help the reader understand this pathway interaction, which is important in the development of cancer.
  2. Page 7, Line 264: I would suggest highlighting that the reduction of cell surface expression of the HLA class I is driven by E5, which leads to tumour escape due to reduced immune control by anti-tumour CD8+ T cells.
  3. Page 11, Paragraph 7: Consider extending this paragraph and including the following points regarding integration:
    1. Partial integration of the HPV genome – episomal DNA vs. integrated DNA.
    2. Integration commonly disrupts the E6 and E7 oncogene regulator, E2, and therefore leads to even more expression of E6 and E7 and drives the carcinogenesis. Consider including this mechanism.

I have found further minor points to change/correct which are listed below:

  1. Abstract, Line 12: It might be worth including E5 in the abstract as well as it also can lead to evasion of host immune responses (e.g. MHC down-regulation).
  2. Figure 1, Legend: Change oncogenic into a lower case “o”: “Natural history of Oncogenic Human Papillomavirus infection”
  3. Figure 2, Legend: You have stated “Green arrows represent the viral open reading frames.”, what is with the purple and light-blue arrows?
  4. Subheading Page 4, Line 123: “4. Overview of the HPV Life cycle”, “cycle” can be written with a capital “C” in the title.
  5. Figure 3, Legend: “lifecycle” is written without space. Please use a consistent way of writing „life cycle“.
  6. Figure 3, Legend: The abbreviation of “PML-NBs” is not explained in the legend. Please include.

Author Response

The authors reviewed and summarized the current literature of the oncogenic human papillomavirus (HPV) types, their related genome structure and gene expression regulation. The authors also depicted the ability of HPV to hide from the immune system and therefore stay persistent for years, which then can lead to cancer development.

The review is presented in an easy-to-read and comprehensible manner. This topic is of (clinical) relevance – not only for virologists - and is in the scope of the journal. It can support the interested reader to understand the HPV pathogenesis in more detail.

However, I would propose extending or editing some paragraphs:

  1. Page 6, Line 15: As E6 and E7 are the key players in cancer development through p53 degradation and cell cycle activation via Rb I would suggest extending this section and consider including a schematic figure explaining those pathways. An illustration could help the reader understand this pathway interaction, which is important in the development of cancer.
  • We have added an illustration (new Figure 5) outlining the ways in which E5, E6 and E7 interact with multiple cellular pathways to promote persistent infection and subsequently lead to cancer development.
  1. Page 7, Line 264: I would suggest highlighting that the reduction of cell surface expression of the HLA class I is driven by E5, which leads to tumour escape due to reduced immune control by anti-tumour CD8+ T cells.
  • We have added more information about the reduction of cell surface expression of the HLA class I driven by E5, as this promotes persistent infection (lines 356-362).
  1. Page 11, Paragraph 7: Consider extending this paragraph and including the following points regarding integration:
    1. Partial integration of the HPV genome – episomal DNA vs. integrated DNA.
    2. Integration commonly disrupts the E6 and E7 oncogene regulator, E2, and therefore leads to even more expression of E6 and E7 and drives the carcinogenesis. Consider including this mechanism.
  • We have added the requested additional information about HPV integration (lines 577-589)

I have found further minor points to change/correct which are listed below:

  1. Abstract, Line 12: It might be worth including E5 in the abstract as well as it also can lead to evasion of host immune responses (e.g. MHC down-regulation).
  2. Figure 1, Legend: Change oncogenic into a lower case “o”: “Natural history of Oncogenic Human Papillomavirus infection”
  3. Figure 2, Legend: You have stated “Green arrows represent the viral open reading frames.”, what is with the purple and light-blue arrows?
  4. Subheading Page 4, Line 123: “4. Overview of the HPV Life cycle”, “cycle” can be written with a capital “C” in the title.
  5. Figure 3, Legend: “lifecycle” is written without space. Please use a consistent way of writing „life cycle“.
  6. Figure 3, Legend: The abbreviation of “PML-NBs” is not explained in the legend. Please include.
  • All Minor points were changed/corrected as recommended.